# Patterns of Beverage Consumption Among Saudi Residents and Associated Demographic Factors: A Nationwide Survey

**DOI:** 10.3390/nu17132182

**Published:** 2025-06-30

**Authors:** Ruyuf Y. Alnafisah, Tahrir M. Aldhirgham, Nouf S. Alammari, Nahlah A. Alhadhrami, Safaa Abdelaziz Alsaaydan, Sarah M. Alamri, Mona Alshamari, Eman Alamri, Majed BinRowiah, Reem Ali Alomari, Amani S. Alqahtani

**Affiliations:** 1Saudi Food and Drug Authority, Riyadh 13513, Saudi Arabia; tmdhirgham@sfda.gov (T.M.A.); nsammari@sfda.gov.sa (N.S.A.); as.qahtani@sfda.gov.sa (A.S.A.); 2Chemistry Department, Faculty of Science, Taibah University, Madinah 42353, Saudi Arabia; nhadhrami@taibahu.edu.sa; 3Department of Clinical Nutrition, College of Nursing and Health Sciences, Jazan University, Jazan 45142, Saudi Arabia; salsaidan@jazanu.edu.sa; 4College of Science, King Khalid University, Abha 61413, Saudi Arabia; dr.sarah.alamri@gmail.com; 5Human Nutrition, Applied College, Hail University, Hail 55473, Saudi Arabia; mona.e.0867@gmail.com; 6Department of Food Science and Nutrition, University of Tabuk, Tabuk 71491, Saudi Arabia; ialamri@ut.edu.sa; 7Independent Researcher, Najran 61441, Saudi Arabia; abnrwiah@gmail.com; 8Department of Nutrition, College of Science, Al-Baha University, Al Bahah 65431, Saudi Arabia; arreen2011@hotmail.com

**Keywords:** beverage, pattern of consumption, energy consumption, total intake

## Abstract

**Background/Objectives**: Non-communicable diseases (NCDs) are strongly linked to beverage consumption. This study aimed to assess the average daily beverage intake of Saudi residents, energy intake from beverages, and the influence of socio-demographic factors, health behaviors, and health status on beverage intake. **Methods**: A nationally representative, cross-sectional study utilized stratified quota sampling to survey adults (≥18 years) across all 13 administrative regions of Saudi Arabia. Data were collected from April 2022 to December 2023 using the validated Arabic Beverage Frequency Questionnaire (ABFQ), assessing consumption patterns of 28 beverage types. **Results**: The study included 4385 participants (mean age: 36.1 ± 11.14 years, 65% male). Sweetened tea (28 mL/day), regular soft drinks (22.1 mL/day), and Saudi coffee (18 mL/day) were the most frequent beverages after water. Sweetened tea contributed to the highest average energy intake (33.2 ± 46.4 kcal/day). Consumption of sugar-sweetened beverages (SSBs) was higher among younger individuals (18–29 years: OR: 4.0, 95% CI [2.6–6.3]; 30–44 years: OR: 2.8, 95% CI [1.8–4.3]), males (OR:1.6, 95% CI [1.4–1.8]), and residents of specific regions [Al-Jawf (OR: 1.9, 95% CI [1.2–3.2]) and Jazan (OR: 3.2, 95% CI [2.2–4.7])]. Higher water intakes were associated with males (OR: 1.5, 95% CI [1.3–1.7]), higher education levels (OR: 1.4, 95% CI [1.2–1.8]), physically active (OR: 1.5, 95% CI [1.3–1.8]), and those overweight (OR: 1.6, 95% CI [1.2–2.3]) or obese (OR: 2, 95% CI [1.4–2.8]). **Conclusions**: This study provides a valuable insight into beverage consumption patterns among Saudi residents. The findings highlight the need for targeted public health interventions to promote healthier beverage choices, particularly among younger populations and those with lower socioeconomic status.

## 1. Introduction

Non-communicable diseases (NCDs), including cardiovascular diseases, cancer, and diabetes, pose a significant global health challenge, accounting for a substantial proportion of all deaths worldwide [1]. The burden of NCDs is similarly alarming in the Middle East and North Africa (MENA) region, where rates of NCD-related mortality mirror global trends [2]. Obesity, a major risk factor for NCDs, is prevalent across the region [3].

Behavioral factors, including poor dietary habits, excessive consumption of sugar-sweetened beverages (SSBs), and physical inactivity, play a critical role in the increasing incidence of these diseases [1]. Among these, beverage consumption has emerged as a key area of concern due to its direct contribution. A growing body of evidence demonstrates the strong association between high intake of SSBs and these health outcomes [4,5,6,7,8,9,10,11,12,13,14,15,16]. Conversely, water consumption is essential for supporting vital physiological functions and overall health, with studies indicating its role in the prevention of chronic diseases [7,17]. The impact of caffeinated beverages, however, is more complex. While they may offer health benefits, their effects can vary, particularly for individuals with pre-existing cardiovascular conditions or hypertension, where both risks and benefits have been observed [18,19].

Studies assessed beverage consumption using food records to determine their contribution to daily energy intake in both adults and children, revealing that plain water is the predominant beverage across all age groups, while other beverages vary by age [20,21,22,23,24,25]. In Saudi Arabia, data on dietary intake, particularly regarding beverage consumption, remain scarce. One study involving 507 undergraduate students reported an average daily intake of 650.6 mL of sugar-sweetened beverages (SSBs), 575.2 mL of caffeinated beverages, and 224.6 mL of carbonated drinks, contributing 187.6 kcal from SSBs, 87.6 kcal from caffeinated beverages, and 52.5 kcal from carbonated drinks, respectively [26].

Despite the critical role of sweet beverage consumption in overall dietary patterns and public health, comprehensive nationwide data in Saudi Arabia are limited. This study aimed to address this gap by providing a comprehensive analysis of beverage consumption patterns among Saudi residents. Specifically, the study investigated the average daily intake of various beverages, estimated energy intake from beverages, and explored the influence of socio-demographic factors, health behaviors, and health status on beverage consumption choices.

## 2. Materials and Methods

### 2.1. Study Design

This study employed a nationally representative, cross-sectional design, using a stratified quota sampling technique to survey adults aged 18 years and older. Data were collected through a validated online questionnaire [27], covering all 13 administrative regions of Saudi Arabia, including Al-Riyadh, Makkah, Al-Madinah, Al-Qaseem, Eastern Province, Aseer, Tabouk, Hail, Northern Borders, Jazan, Najran, Al-Baha, and Aljouf. The study was reviewed and approved by the SFDA Ethical Committee (Approval number: 2022_1).

### 2.2. Sample Size

A multistage quota sampling technique was used to achieve equal distribution of participants across the 13 regions of Saudi Arabia stratified by region, age, and gender. Two age groups were used based on the median age of adults in Saudi Arabia, reported as 37 in the latest census data [28]. This led to 52 quotas for this study. The sample size was calculated based on the depth of the sub-analysis we wanted to reach, which is comparing the age and gender groups between regions with a medium effect size of around 0.3. Thus, each quota required 68 participants, resulting in a total sample size of 3536 participants. Once the quota sample is reached, the participants with similar characteristics will not be eligible to participate in the study. All statistics of this survey have been calculated using sample weights assigned to each demographic quota that were constructed to represent the Saudi population [28]. The proportions of the population within each demographic quota were calculated to reflect their representation in the overall population. For each stratum, sample proportions were determined based on the collected data. Sample weights were then derived by dividing the population proportion of each quota by its corresponding sample proportion.

### 2.3. Sample Recruitment

Participants were included in the study if they met the following criteria: (a) residents of Saudi Arabia (both Saudis and non-Saudis); (b) aged 18 years and above; and (c) Arabic-speaking. A list of random phone numbers was generated from a government database based on the study’s inclusion criteria. A link to the survey was then sent via Short Message Service (SMS), including study objectives, a consent form regarding the confidentiality and anonymity of personal information, followed by the eligibility questions. Data collection was conducted from April 2022 until December 2023. However, due to major changes in dietary behavior in Ramadan (i.e., the Muslim fasting month), data collection was postponed during this period.

### 2.4. Measurements and Variable Definitions

#### 2.4.1. Arabic Beverage Frequency Questionnaire (ABFQ)

The ABFQ is a validated online tool that measures total beverage intake among Arabic-speaking consumers [27]. The questionnaire included an assessment of consumption of 28 beverages: water, unpackaged 100% fresh fruit juice, unpackaged 100% fresh vegetable juice, packaged 100% fresh fruit juice, packaged fruit drinks or nectars, packaged fruit juice with milk, full-fat milk, low-fat milk, skimmed-fat milk, full-fat buttermilk (Laban), low-fat buttermilk (Laban), skimmed-fat buttermilk (Laban), regular soft drinks, diet soft drinks, packaged iced tea, unsweetened tea, sweetened tea, Saudi coffee, Saudi coffee with milk, unsweetened Turkish coffee or espresso, sweetened Turkish coffee or espresso, unsweetened black or filtered coffee, black or filtered coffee with milk, flavored coffee with milk such as cappuccino, flavored malt drink, sweetened energy drink, diet energy drink, and sports drinks.

To evaluate beverage consumption, participants were queried regarding their drinking patterns over the preceding 30 days. Responses were classified into 11 distinct consumption frequency levels, which were subsequently transformed into daily frequency units. These units were defined as follows: never (0), once a month (0.033), twice a month (0.067), three times a month (0.1), once a week (0.14), two to three times a week (0.36, averaged at 2.5 days), four to six times a week (0.71, averaged at 5 days), once a day (1), twice a day (2), three times a day (3), and more than three times a day (4). To assess the quantity consumed, participants were asked about the usual consumption amount each time. The frequency codes established were then multiplied by the reported amount consumed (in mL) per occasion to calculate the average daily intake in mL.

To calculate energy intake from beverages, the energy content of each beverage was first determined. Beverages were categorized into two groups: (1) pre-packaged beverages, for which the average energy content per 100 mL was derived from the Saudi Branded Food Database (SBFD) [29], and (2) unpackaged beverages, where energy content per 100 mL was obtained from the ESHA Food Database [30]. Once the energy content for each beverage was established, the daily energy intake was calculated by multiplying the daily volume consumed (in milliliters) by the energy content per 100 mL of that particular beverage and then dividing by 100. Outliers for all continuous variables were then detected using box plots and replaced with the median value for the corresponding variable [31].

For analysis, beverages were categorized into five groups: (1) water; (2) 100% fresh fruits and vegetable juices, including unpackaged 100% fresh fruit juice, unpackaged 100% fresh vegetable juice, and packaged 100% fresh fruit juice; (3) sugary-sweetened beverages (SSB), including packaged fruit drinks or nectars, regular soft drinks, diet soft drinks, packaged iced tea, flavored malt drink, sweetened energy drink, diet energy drink, and sport drink; (4) dairy beverages, including packaged fruit juice with milk, full-fat milk, low-fat milk, skimmed-fat milk, full-fat buttermilk (Laban), low-fat buttermilk (Laban), and skimmed-fat buttermilk (Laban); and (5) caffeinated beverages, including unsweetened tea, sweetened tea, Saudi coffee, Saudi coffee with milk, unsweetened Turkish coffee or espresso, sweetened Turkish coffee or espresso, unsweetened black or filtered coffee, black or filtered coffee with milk, and flavored coffee with milk such as cappuccino.

#### 2.4.2. Information on General Characteristics 

The general characteristics section collected key information about socio-demographic factors, health behaviors, and health status. Socio-demographic data, including age, gender, administrative region of residence, educational level (categorized as high school or less, diploma or bachelor’s degree, and postgraduate), and marital status (single, married, and divorced/widowed). Age was classified into four groups according to the World Health Organization (WHO): 18–29, 30–44, 45–59, and 60 years and older [32].

Health behaviors were evaluated by asking participants about their engagement in regular physical activity over the past three months (regular physical activity), as well as their consumption patterns of fruits and vegetables (never, monthly, weekly, and daily). Health status included self-reported weight and height, which were used to calculate body mass index (BMI), as well as inquiries regarding participants’ histories of chronic diseases and any health conditions requiring fluid restriction in the past month. BMI was classified into four groups according to WHO standards: underweight (<18.5 kg/m^2^), normal weight (18.5 to 24.9 kg/m^2^), overweight (25.0 to 29.9 kg/m^2^), and obese (>30 kg/m^2^) [33].

### 2.5. Statistical Analysis

Data analysis was performed using SPSS version 29. General characteristics of the sample were summarized using frequencies and percentages. Daily beverage intake (mL) and energy consumption from beverages (kcal) were described using medians and ranges as the primary measures of central tendency and variability due to the non-normal distribution of the data, which was confirmed by the Shapiro–Wilk test and examination of skewness. Means and standard deviations (SDs) were also reported for reference. Beverage consumption frequencies were categorized as never, daily, weekly, or monthly, and reported with frequencies and percentages. Differences in mean daily beverage intake across various factors, including socio-demographic characteristics, health behaviors, and health status, were evaluated using Independent *t*-tests and One-Way ANOVA, as appropriate. Statistically significant independent factors were included in a binary logistic regression analysis to examine their association with daily beverage intake (in mL) for each beverage category. The dependent variable (daily beverage intake) was categorized based on the median value into two groups: below the median and at or above the median.

## 3. Results

### 3.1. General Characteristics

The sample was predominantly male (65.2%) with a mean age of 36.1 ± 11.1 years. Most participants were physically inactive, and nearly half consumed fruits and vegetables weekly. Overweight and obesity were common, affecting over 60% of the sample. Additionally, about one-fifth reported chronic diseases or conditions requiring fluid restriction. Further details are available in Table 1.

### 3.2. Beverages Total Intake

Table 2 summarizes the median daily intake of 28 beverage categories. Water was the most consumed beverage, followed by sweetened tea, regular soft drinks, and Saudi coffee. More details are presented in Table 2.

### 3.3. Energy Consumption

Table 3 presents the median estimated energy intake from various beverages. Regular soft drinks and sweetened tea contributed the highest median energy, followed by full-fat buttermilk and fresh fruit juices. Several beverages, including black coffee with milk and skimmed-fat dairy drinks, had median energy intakes near zero despite wide variability. Full details are available in Table 3.

### 3.4. Pattern of Consumption

Distinct consumption patterns were observed for the 28 beverages examined (Table 4). Saudi coffee was most frequently consumed on a daily basis. In contrast, unpackaged 100% fresh fruit juices were predominantly consumed monthly.

### 3.5. Factors Associated with Beverages Total Intake

#### 3.5.1. Socio-Demographic Factors

Table 5 shows socio-demographic factors associated with beverage consumption. Younger adults (18–44 years) were more likely to consume sugar-sweetened beverages compared to older adults. Males had higher odds of consuming water, 100% fresh fruit and vegetable juices, and sugar-sweetened beverages than females. Regional differences were observed in the consumption of dairy and caffeinated beverages, with several regions showing higher intakes compared to Riyadh. Higher educational attainment was linked to increased water consumption but lower intake of sugar-sweetened beverages. Marital status also influenced beverage choices, with married individuals less likely to drink water but more likely to consume caffeinated beverages. Further details are provided in Table 5.

#### 3.5.2. Health Behaviors

Table 6 presents associations between health behaviors and beverage consumption. Participants who engaged in regular physical activity were more likely to consume water and dairy beverages. Additionally, daily or weekly fruit and vegetable intake was significantly linked to higher consumption of water, 100% fresh fruit and vegetable juices, and dairy beverages. Detailed results can be found in Table 6.

#### 3.5.3. Health Status

Table 7 highlights associations between health status and beverage consumption. Overweight and obese participants were more likely to consume water compared to underweight individuals. Those without chronic diseases or fluid restriction conditions had higher odds of consuming sugar-sweetened beverages, dairy drinks, and 100% fresh fruit and vegetable juices. Further details are available in Table 7.

## 4. Discussion

This study provides valuable insights into the beverage consumption patterns of Saudi residents. Our findings demonstrate that water is the most frequently consumed beverage, followed by sweetened tea, regular soft drinks, and traditional Saudi coffee. Notably, regular soft drinks emerged as the primary contributor to energy intake from beverages, highlighting the potential health implications of high sugar consumption. Furthermore, the study underscores the significant influence of socio-demographic factors, health behaviors, and health status on beverage choices. These findings have crucial implications for public health, providing a foundation for the development and implementation of targeted interventions aimed at promoting healthier beverage consumption among the Saudi population.

Sweetened tea ranked as the second most consumed beverage. A cross-sectional study among Saudi adolescents reported that 19.6% of participants regularly consumed sugar-sweetened tea, underscoring its significant contribution to total energy intake [34]. Similarly, findings from the National Health and Nutrition Examination Survey (NHANES) 2007–2010 revealed a positive association between higher sugar-sweetened beverage (SSB) intake, including sweetened tea, and increased overall caloric consumption [35]. However, findings in SSB consumption are not uniform across populations; a study assessing beverage consumption patterns documented a decline in sweetened tea and powdered drink intakes between 2018 and 2019 [36]. Despite these variations, the persistently high intake of sweetened tea highlights its role in excessive energy consumption, which may elevate the risk of metabolic disorders such as obesity and type 2 diabetes [6]. This finding likely reflects cultural preferences, habitual consumption patterns, and product accessibility, emphasizing the need for public health interventions to encourage healthier beverage choices.

The findings indicate that regular soft drinks are among the most consumed beverages among Saudi residents. This high consumption rate raises significant health concerns due to its strong association with obesity, diabetes, and other metabolic disorders linked to high sugar content [12,37,38]. The high intake may reflect cultural or lifestyle preferences for sugary beverages [39]. In response to these concerns, the Saudi government implemented a 50% excise tax on carbonated soft drinks to discourage consumption by increasing prices from 2.31 to 2.5 Saudi Riyals per pack, resulting in a 31% decline in volume sales [40,41]. Despite these efforts, high consumption persists, indicating a need for further regulatory actions. Promoting the substitution of soft drinks with healthier alternatives, such as water or unsweetened beverages, could significantly alleviate the negative health outcomes associated with excessive sugar intake [42,43,44].

The findings indicate that Saudi coffee, or locally known as Gahwa, is one of the most frequently consumed beverages in Saudi Arabia, reflecting its deep cultural significance and role in social customs [45]. Its popularity arises from both its flavor and its association with hospitality and social gatherings [45]. Although typically prepared with less sugar than other beverages, Saudi coffee is often consumed with dates or sweet snacks, which can increase overall sugar intake [46]. Additionally, Saudi coffee is typically served in small cups and consumed multiple times throughout the day, which may result in a cumulative increase in caffeine intake [47]. These findings underscore the importance of understanding cultural dietary habits. Although Saudi coffee offers health benefits due to its antioxidant properties [48], promoting moderation and awareness of excessive caffeine consumption is crucial.

The findings of this study reveal a significant finding among younger adults, particularly those aged 18–29 and 30–44 years, who demonstrate a stronger preference for SSBs compared to individuals aged 60 and older. This observation aligns with the existing literature, highlighting the variations in beverage consumption patterns across different age groups [49,50,51]. Several factors contribute to the higher consumption of SSBs among younger demographics, including taste preferences, increased exposure to aggressive SSB marketing, and the beverages’ wide availability and affordability [51,52]. Additionally, these younger individuals may be more susceptible to developing dependence on these sugary drinks [51,52]. Understanding these patterns is essential for designing targeted public health interventions aimed at reducing SSB consumption among this age group. A multi-pronged approach is needed to address this challenge, including strategies such as public education campaigns, stricter regulations on SSB advertising, particularly to younger audiences, and promoting healthier alternatives.

This study reveals significant regional variations in beverage consumption patterns, which can be driven by interrelated factors such as environmental conditions and cultural practices. For instance, residents of Makkah exhibited a greater likelihood of consuming water compared to those in Riyadh, which can be attributed to the hotter climate of the Makkah region that necessitates increased hydration [53,54,55]. In contrast, the southern regions of Saudi Arabia—specifically Jazan, Najran, and Asir—show a higher prevalence of dairy beverage consumption. This finding is supported by local agricultural practices, particularly livestock farming, which facilitates the availability of fresh milk and dairy [56]. In Qassim, Hail, Aljawf, and Jazan, residents display a notable preference for caffeinated beverages, reflecting the cultural significance of coffee, especially Saudi coffee [45,57,58,59]. Understanding these regional variations is crucial for developing culturally relevant public health initiatives that encourage healthier beverage choices.

The current study found that individuals with higher educational attainment were significantly more likely to consume water and less likely to consume SSBs compared to those with lower levels of education. This pattern is consistent with previous research indicating that health-conscious behaviors tend to be more prevalent among populations with higher educational backgrounds [60,61]. This relationship may be attributed to education’s role in fostering greater health awareness, which is linked to healthier lifestyle choices [62,63,64,65]. Moreover, individuals with higher education levels often have better access to health-promoting resources, such as nutritional information [66]. Understanding these dynamics can inform public health initiatives aimed at promoting healthier drinking habits, particularly among lower-educated populations.

The study revealed that males, physically active individuals, and those classified as overweight or obese had higher water consumption. This can be explained by differences in metabolism and physiological needs. Men generally have a higher muscle mass than women, leading to increased water requirements for maintaining cellular hydration and metabolic processes [67]. Similarly, physically active individuals require more water to compensate for fluid lost through sweat during exercise, as well as to support enhanced metabolic activity during physical exertion [68]. Overweight and obese individuals often have higher metabolic rates and increased body mass, which necessitate greater fluid intake to regulate body temperature and maintain metabolic functions [69,70].

This study has several limitations that should be acknowledged, including its cross-sectional design, which provides observations on beverage consumption frequencies and self-reported health status at a single point in time, without assessing clinical health outcomes. Health status was self-reported and not verified through medical records, which may introduce reporting bias. Additionally, while energy intake from beverages was estimated based on reported consumption and standard energy values, total daily energy intake was not directly measured. The reliance on self-reported data may introduce recall bias, leading participants to inaccurately estimate their beverage intake. Nevertheless, the study’s strengths include the use of a comprehensive online cross-sectional survey design, which allowed for the collection of data from a large and diverse sample. The use of the validated ABFQ also provides a reliable tool for measuring beverage intake among Arabic-speaking populations. Furthermore, all statistics of this survey were calculated using sample weights assigned to each demographic quota constructed to represent the Saudi population. The findings hold important implications for public health strategies aimed at promoting healthier beverage consumption among Saudi residents while addressing the need to mitigate the intake of sugary beverages.

## 5. Conclusions

This study provides a comprehensive assessment of beverage consumption patterns among Saudi residents, highlighting key socio-demographic and health factors associated with beverage choices. The findings emphasize the prevalence of sugar-sweetened beverages among young age groups and the role of education in promoting healthier beverage choices. This study serves as a crucial step in understanding beverage consumption behaviors in Saudi Arabia, providing a foundation for targeted public health initiatives aimed at reducing the consumption of sugary beverages and encouraging healthier alternatives to improve overall health outcomes.

## Figures and Tables

**Table 1 nutrients-17-02182-t001:** General characteristics of the study sample.

Variables		Unweighted—n (%)	Weighted—n (%) *
**Overall**		**4385 (100%)**	**4385 (100%)**
Socio-demographics			
Age—Mean (SD)		36.1 (11.3)	36.1 (11.1)
Age	18–29	1339 (30.5%)	1097 (31%)
	30–44	2042 (46.5%)	1712 (48.4%)
	45–59	911 (20.7%)	641 (18.1%)
	+60	104 (2.4%)	88 (2.5%)
Gender	Male	1992 (45.3%)	2861 (65.2%)
	Female	2404 (54.7%)	1525 (34.8%)
Regions	Riyadh	492 (11.2%)	1217 (27.8%)
	Eastern	327 (7.4%)	711 (16.2%)
	Al Bahah	398 (9.1%)	44 (1%)
	Al-Jawf	291 (6.6%)	73 (1.7%)
	Northern border	190 (4.3%)	45 (1%)
	Najran	312 (7.1%)	72 (1.6%)
	Qassim	265 (6%)	177 (4%)
	Medina	483 (11%)	281 (6.4%)
	Tabuk	308 (7%)	112 (2.6%)
	Jazan	309 (7%)	176 (4%)
	Hail	305 (6.9%)	97 (2.2%)
	Asir	363 (8.3%)	262 (6%)
	Makkah	353 (8%)	1119 (25.5%)
Education	High school and less	1045 (23.8%)	991 (22.6%)
	Diploma or bachelor	2806 (63.8%)	2758 (62.9%)
	Post-graduate	545 (12.4%)	637 (14.5%)
Marital status	Single	1381 (31.4%)	1456 (33.2%)
	Married	2771 (63%)	2755 (62.8%)
	Divorced/widow	244 (5.6%)	175 (4%)
Health behaviors			
Regular physical activity	Yes	1175 (26.7%)	1288 (29.4%)
	No	3221 (73.3%)	3098 (70.6%)
Fruit and vegetable consumption	Never	94 (2.1%)	87 (2%)
	Monthly	1104 (25.1%)	1113 (25.4%)
	Weekly	2042 (46.5%)	2116 (48.2%)
	Daily	1156 (26.3%)	1070 (24.4%)
Health status			
BMI	Underweight	204 (4.6%)	163 (3.7%)
	Normal weight	1502 (34.2%)	1475 (33.6%)
	Overweight	1664 (37.9%)	1712 (39%)
	Obese	1026 (23.3%)	1036 (23.6%)
Presence of chronic diseases (Yes)		958 (21.8%)	1013 (23.1%)
Cardiovascular diseases such as stroke and atherosclerosis		62 (1.4%)	74 (1.7%)
Hypertension		248 (5.6%)	256 (5.8%)
Diabetes		304 (6.9%)	335 (7.6%)
Respiratory diseases such as asthma and chronic obstructive pulmonary disease		127 (2.9%)	145 (3.3%)
Cancer		21 (0.5%)	26 (0.6%)
Osteoporosis		47 (1.1%)	30 (0.7%)
Alzheimer		5 (0.1%)	6 (0.1%)
Other chronic diseases		343 (7.8%)	382 (8.7%)
Presence of a health condition that requires fluid restriction during the last month (Yes)		901 (20.5%)	846 (19.3%)
Arthritis		112 (2.5%)	128 (2.9%)
Heart diseases, such as heart failure		58 (1.3%)	61 (1.4%)
Internal disease or constipation		298 (6.8%)	271 (6.2%)
Surgery		45 (1%)	50 (1.1%)
Kidney disease, such as kidney failure and dialysis		14 (0.3%)	22 (0.5%)
Cold or gastrointestinal illness or diarrhea		226 (5.1%)	215 (4.9%)
Urinary tract infection/kidney stones		183 (4.2%)	150 (3.4%)
Other health conditions that require fluid restriction during the last month		168 (3.8%)	169 (3.9%)

* Data are weighted to represent the Saudi population based on quotas for region, age, and gender.

**Table 2 nutrients-17-02182-t002:** Average daily intake (mL/day) per person.

Beverages	mL/Day	
	Median (Range Width)	Mean (SD)
Water	990 (3200)	1147.2 (687)
Sweetened tea	28 (455)	80.89 (113.3)
Regular soft drinks	22.1 (284)	45.9 (65.7)
Saudi coffee	18 (240)	39.1 (50.8)
Unpackaged 100% fresh fruit juice	15.7 (90)	17.9 (19.6)
Full-fat buttermilk (Laban)	13.4 (144)	20.8 (28.9)
Full-fat milk	9.075 (200)	29.2 (49.4)
Packaged 100% fresh fruit juice	7.755 (64.8)	9.4 (12.2)
Black or filtered coffee with milk	0 (2400)	22.12 (102.6)
Packaged iced tea	0 (1065)	18.5 (78.3)
Sweetened energy drink	0 (2000)	18.4 (78.6)
Unsweetened tea	0 (144)	15.5 (34.8)
Skimmed-fat milk	0 (1100)	13.6 (72.2)
Skimmed-fat buttermilk (Laban)	0 (1080)	11 (65.7)
Diet energy drink	0 (1200)	9.6 (58.9)
Unsweetened black or filtered coffee	0 (99.4)	7 (18.8)
Flavored coffee with milk, like cappuccino	0 (49.7)	6.8 (12.2)
Packaged fruit drinks or nectars	0 (44.1)	6.7 (10)
Sport drink	0 (1100)	6.3 (45.8)
Diet soft drinks	0 (28)	2.2 (5.7)
Low-fat milk	0 (20)	1.4 (4)
Unsweetened Turkish coffee or espresso	0 (16.8)	1.1 (2.7)
Flavored malt drink	0 (15.7)	1.1 (3.2)
Unpackaged 100% fresh vegetable juice	0 (15.7)	0.9 (2.8)
Packaged fruit juice with milk	0 (14.9)	0.8 (2.5)
Low-fat buttermilk (Laban)	0 (13.4)	0.7 (2.4)
Saudi coffee with milk	0 (7)	0.4 (1.4)
Sweetened Turkish coffee or espresso	0 (0)	0 (0)

**Table 3 nutrients-17-02182-t003:** Average energy consumption (kcal/day) from beverages per person.

Beverages	kcal/day	
	Median (Range Width)	Mean (SD)
Regular soft drinks	12 (154)	25 (36)
Sweetened tea	11 (187)	33 (46)
Full-fat buttermilk (Laban)	8 (85)	12 (17)
Unpackaged 100% fresh fruit juice	8 (45)	9 (10)
Full-fat milk	6 (125)	18 (31)
Packaged 100% fresh fruit juice	4 (30)	4 (6)
Black or filtered coffee with milk	0 (304)	3 (13)
Skimmed-fat milk	0 (571)	7 (37)
Packaged iced tea	0 (380)	7 (28)
Skimmed-fat buttermilk (Laban)	0 (411)	4 (25)

Note: Beverages contributing less than 1 kcal/day on average, including unsweetened tea, unsweetened black or filtered coffee, water, diet soft drinks, and other low-energy drinks, are not individually listed due to their negligible energy contribution.

**Table 4 nutrients-17-02182-t004:** Beverage consumption patterns.

Beverages	Pattern of Consumption—n (%)			
	Never	Monthly	Weekly	Daily
Unpackaged 100% fresh fruit juice	1189 (27.1%)	2018 (46%)	903 (20.6%)	275 (6.3%)
Unpackaged 100% fresh vegetable juice	3369 (76.8%)	649 (14.8%)	254 (5.8%)	114 (2.6%)
Packaged 100% fresh fruit juice	1827 (41.7%)	1548 (35.3%)	768 (17.5%)	243 (5.5%)
Packaged fruit drinks or nectars	2238 (51%)	1229 (28%)	707 (16.1%)	211 (4.8%)
Packaged fruit juice with milk	3517 (80.2%)	584 (13.3%)	179 (4.1%)	107 (2.4%)
Full-fat milk	1692 (38.6%)	1199 (27.3%)	961 (21.9%)	534 (12.2%)
Low-fat milk	3174 (72.4%)	617 (14.1%)	384 (8.8%)	211 (4.8%)
Skimmed-fat milk	3829 (87.3%)	267 (6.1%)	190 (4.3%)	98 (2.2%)
Full-fat buttermilk (Laban)	1431 (32.6%)	1410 (32.2%)	1100 (25.1%)	444 (10.1%)
Low-fat buttermilk (Laban)	3315 (75.6%)	563 (12.8%)	348 (7.9%)	160 (3.6%)
Skimmed-fat buttermilk (Laban)	3958 (90.2%)	209 (4.8%)	130 (3%)	88 (2%)
Regular soft drinks	1165 (26.6%)	1277 (29.1%)	1270 (29%)	673 (15.3%)
Diet soft drinks	2854 (65.1%)	743 (17%)	562 (12.8%)	226 (5.1%)
Packaged iced tea	3450 (78.7%)	567 (12.9%)	280 (6.4%)	89 (2%)
Unsweetened tea	2280 (52%)	637 (14.5%)	720 (16.4%)	749 (17.1%)
Sweetened tea	1465 (33.4%)	658 (15%)	929 (21.2%)	1335 (30.4%)
Saudi coffee	1022 (23.3%)	751 (17.1%)	1101 (25.1%)	1512 (34.5%)
Saudi coffee with milk	3207 (73.1%)	386 (8.8%)	411 (9.4%)	381 (8.7%)
Unsweetened Turkish coffee or espresso	2590 (59.1%)	708 (16.1%)	537 (12.2%)	550 (12.6%)
Sweetened Turkish coffee or espresso	3445 (78.5%)	497 (11.3%)	253 (5.8%)	191 (4.4%)
Unsweetened black or filtered coffee	2456 (56%)	605 (13.8%)	586 (13.4%)	739 (16.9%)
Black or filtered coffee with milk	3684 (84%)	339 (7.7%)	224 (5.1%)	139 (3.2%)
Flavored coffee with milk, like cappuccino	2304 (52.5%)	1197 (27.3%)	628 (14.3%)	258 (5.9%)
Flavored malt drink	3066 (69.9%)	884 (20.2%)	337 (7.7%)	99 (2.3%)
Sweetened energy drink	3333 (76%)	637 (14.5%)	302 (6.9%)	113 (2.6%)
Diet energy drink	3947 (90%)	241 (5.5%)	137 (3.1%)	60 (1.4%)
Sport drink	4137 (94.3%)	115 (2.6%)	91 (2.1%)	42 (1%)

**Table 5 nutrients-17-02182-t005:** Binary logistic regression between total beverage intake and socio-demographic factors (<median vs. ≥median).

Characteristics		Water		100% Fresh Fruits and Vegetable Juices		Packaged Sugary- Sweetened Beverages (SSB)		Dairy Beverages		Caffeinated Beverages	
		OR	95% CI	OR	95% CI	OR	95% CI	OR	95% CI	OR	95% CI
Age (reference: +60)	18–29	1.1	0.7–1.7	-	-	**4**	**2.6–6.3**	0.8	0.5–1.1	1.1	0.7–1.6
	30–44	1.0	0.7–1.5	-	-	**2.8**	**1.8–4.3**	0.9	0.6–1.3	1.2	0.9–1.8
	45–59	1.3	0.8–1.9	-	-	1.4	0.9–2.2	0.8	0.5–1.1	1	0.7–1.5
Gender (reference: Female)	Male	**1.5**	**1.3–1.7**	**1.4**	**1.2–1.5**	**1.6**	**1.4–1.8**	-	-	1.2	1–1.3
Region (reference: Riyadh)	Eastern Province	1.2	1–1.4	0.9	0.7–1	0.9	0.7–1.1	1.1	0.9–1.3	1	0.8–1.2
	Al Bahah region	1.2	0.6–2.3	1	0.5–1.8	1.2	0.6–2.3	1.6	0.8–2.9	0.9	0.5–1.7
	Al-Jawf region	0.7	0.4–1.1	1	0.6–1.7	**1.9**	**1.2–3.2**	**2**	**1.2–3.3**	**2**	**1.2–3.3**
	Northern border region	0.7	0.4–1.3	1	0.5–1.9	0.9	0.5–1.6	1	0.6–1.9	1.4	0.8–2.6
	Najran region	0.7	0.4–1.1	1.1	0.7–1.8	1.1	0.7–1.8	**1.7**	**1.1–2.8**	1.5	0.9–2.4
	Qassim region	**0.7**	**0.5–0.9**	0.5	0.4–0.8	0.9	0.6–1.2	1.4	1–1.9	**1.6**	**1.1–2.2**
	Medina region	1	0.7–1.3	1	0.8–1.3	0.9	0.7–1.2	1.3	1–1.6	1	0.7–1.2
	Tabuk region	0.8	0.5–1.2	0.8	0.6–1.2	0.9	0.6–1.4	1.5	1–2.2	1.3	0.8–1.9
	Jazan region	**0.6**	**0.5–0.9**	**2.6**	**1.8–3.7**	**3.2**	**2.2–4.7**	**3.6**	**2.5–5.2**	**2.2**	**1.6–3**
	Hail region	**0.4**	**0.3–0.6**	0.8	0.5–1.2	1	0.7–1.6	1.1	0.7–1.7	**1.7**	**1.1–2.6**
	Asir region	**0.6**	**0.5–0.8**	0.8	0.6–1	1.2	0.9–1.5	**1.7**	**1.3–2.2**	1.3	1–1.6
	Makkah Region	**1.4**	**1.1–1.6**	0.8	0.7–1	1	0.8–1.1	1	0.9–1.2	1	0.8–1.2
Education (reference: High school and less)	Diploma or bachelor	**1.4**	**1.2–1.6**	-	-	**0.7**	**0.6–0.9**	1	0.8–1.1	-	-
	Post-graduate	**1.4**	**1.2–1.8**	-	-	**0.6**	**0.5–0.7**	0.9	0.8–1.2	-	-
Marital status (reference: Single)	Married	**0.6**	**0.5–0.8**	1	0.9–1.2	0.9	0.7–1	-	-	**1.3**	**1.1–1.5**
	Divorced/widow	0.7	0.5–1	1.3	0.9–1.8	1.1	0.7–1.5	-	-	**1.7**	**1.2–2.4**

Note: Bolded values indicate statistically significant results.

**Table 6 nutrients-17-02182-t006:** Binary logistic regression between total beverage intake and health behaviors (<median vs. ≥median).

Characteristics		Water		100% Fresh Fruits and Vegetable Juices		Packaged Sugary-Sweetened Beverages (SSB)		Dairy Beverages		Caffeinated Beverages	
		OR	95% CI	OR	95% CI	OR	95% CI	OR	95% CI	OR	95% CI
Practicing physical activity regularly for 3 months or more (reference: No)	Yes	**1.5**	**1.3–1.8**	-	-	0.8	0.7–1	**1.3**	**1.1–1.5**	0.9	0.8–1
Fruit and vegetable consumption (reference: Never)	monthly	1.5	0.9–2.3	1.5	0.9–2.5	1.4	0.9–2.2	**1.9**	**1.1–3.1**	-	-
	weekly	**2.1**	**1.3–3.3**	**3.6**	**2.2–5.9**	1.2	0.8–1.9	**2.8**	**1.7–4.5**	-	-
	daily	**2.6**	**1.6–4.1**	**4.2**	**2.6–7**	1.2	0.7–1.7	**3.1**	**1.9–5.1**	-	-

Note: Bolded values indicate statistically significant results.

**Table 7 nutrients-17-02182-t007:** Binary logistic regression between total beverage intake and health status (<median vs. ≥median).

Characteristics		Water		100% Fresh Fruits and Vegetable Juices		Packaged Sugary-Sweetened Beverages (SSB)		Dairy Beverages		Caffeinated Beverages	
		OR	95% CI	OR	95% CI	OR	95% CI	OR	95% CI	OR	95% CI
BMI (reference: Underweight)	Normal weight	1.2	0.8–1.7	1.3	0.9–1.9	1.1	0.8–1.5	-	-	-	-
	Overweight	**1.6**	**1.2–2.3**	1.3	0.9–1.8	1	0.7–1.4	-	-	-	-
	Obese	**2**	**1.4–2.8**	1	0.7–1.4	1.1	0.8–1.6	-	-	-	-
Presence of chronic diseases (reference: Yes)	No	0.9	0.7–1	-	-	**1.3**	**1.1–1.5**	**1.2**	**1.1–1.4**	-	-
Presence of a health condition that requires fluid restriction during the last month (reference: Yes)	No	-	-	**1.3**	**1.1–1.5**	1	0.8–1.2	-	-	-	-

Note: Bolded values indicate statistically significant results.

## Data Availability

Data available on request. The data are not publicly available due to Privacy.

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
