# Peer review of "Patterns of Beverage Consumption Among Saudi Residents and Associated Demographic Factors: A Nationwide Survey"

_nutrients, 2025, doi:10.3390/nu17132182_

Round 1

Reviewer 1 Report

Comments and Suggestions for Authors

Alnafisah RY et al's manuscript showed that sweetened tea was the largest contributor to energy intake from beverages and that excessive sugar intake poses a potential risk to the development of non-communicable diseases among young age groups of Saudi residents. 

In Table1, there is a characteristics of Health status. Can the authors show the relationship between nutritional status and disease? The subjects were classified into four groups based on BMI, but I would like to know if the distribution of diseases in each group was related to subsequent beverage preferences. I would also like to discuss the correlation between the severity of diseases and the cases of people suffering from multiple diseases.

In Table 2, is the total intake of beverage per person? It's not specified. Can the authors show it by gender or age?

There are no graphs or tables to illustrate section 3.5.

Since this is an international journal, why not take a closer look at how the actual situation in Saudi Arabia revealed through this survey compares with other regions?

I thought there was little relationship between beverage intake and NCDs.

Minor points,

Page4 Line163, WHO standards: under weight (<18.5 kg/m2)・・

Page6 Line191, Beverage total intake (ml/day

Page7 Line 208, form → from

Reviewer 2 Report

Comments and Suggestions for Authors

The manuscript entitled “Patterns of Beverage Consumption Among Saudi Residents and  Associated Demographic Factors: A Nationwide Survey” presents interesting issues however some questions arise.

  • Lines 64-65 – “Despite the critical role of beverage consumption in overall dietary patterns and public health, comprehensive nationwide data in Saudi Arabia are limited” – The term 'sweet beverage' seems more appropriate than simply 'beverage’.

  • Line 68 – “[…] assessed their contribution to total energy intake […] ” - How did the authors assess the contribution of energy from daily intake of beverages to total energy intake, if there is no information in the article regarding total energy intake? Please revise, clarify, or add this information."

  • Lines 133-135 –“ Once the energy content for each beverage was established, daily energy intake was calculated by multiplying the daily beverage intake by its respective energy content and dividing by 100.” – Please clarify this point, as the sentence is unclear—what exactly is meant by 'respective energy content'?

  • Lines 155-137 – “Outliers for all continuous variables were then detected using boxplots, and were replaced with the median value for the corresponding variable.” – Additionally, please provide a citation or justification for replacing outliers with the median value instead of removing them. Why was this strategy chosen, and was the normality of the data distribution analyzed?

  • I do not understand the notation in Table 2 — if it says 'median (range)', why is there only a single value instead of a range? Please clarify this precisely.

  • Table 3 – Unless there is a calculation error (which I asked about above), my question concerns the description in the methodology — the contribution of sweet beverages to daily energy consumption appears minimal, at the level of, for example, 33 kcal/day (Sweetened tea) or 24 kcal /day (Regular soft drinks). Depending on how substantial the correction to the results is, the interpretation of the findings will also need to be adjusted accordingly. At this stage, it is difficult to fully support the authors' statement in the discussion, namely that "sweetened tea emerged as the primary contributor to energy intake from beverages, highlighting the potential health implications of high sugar consumption."

  • This is a cross-sectional study (at a single point in time), therefore it is not appropriate to discuss trends in the discussion section, such as suggesting that younger groups do one thing and older groups do another as a tendency. These are merely observations at one point in time, and the language should be adjusted accordingly.

  • Lines 334- 336 – “This study has several limitations that should be acknowledged, including its cross-sectional design, which restricts the ability to infer causal relationships between beverage consumption and health outcomes.” – The limitations section needs to be revised — no health outcomes were almost not analyzed, so this cannot be discussed in such perspective. This is simply a study on the frequency of beverage consumption and declaration of health status. In the current manuscript, it is even difficult to determine whether the percentage contribution to total daily energy intake was actually assessed. The results and their implications must be described precisely. The conclusions are likely overstated. They are too strongly worded in relation to the type and quality of the study conducted — we only have self-reported health status without verification from medical records — and in light of the statistical analyses performed.

  • Lines 244-245 – “ Overweight (OR: 1.6, 95% CI [1.2–2.3]) and obese participants (OR: 2, 95% CI [1.4–245 2.8]) were more likely to drink water than underweight individuals. - This finding is not explained in the discussion and appears to contradict the authors' own conclusions. Considering the following sentence, "Our findings demonstrate that water is the most frequently consumed beverage...," the conclusions presented do not clearly follow from the results obtained.

Round 2

Reviewer 1 Report

Comments and Suggestions for Authors

Alnafisah et al.'s revised manuscript have been altered as the reviewers suggested.

Table 5 showed the most important results and is essential in this manuscript. 

Table 5 had too much information. As the section 3.5, can authors split Table 5 into 3 or more tables with each factor?

Reviewer 2 Report

Comments and Suggestions for Authors

Thank you very much to the authors for their responses. However, based on the replies received, I now have even more questions. These issues require clarification and most likely some corrections in the article.

In response to the question about the methodology used for the outliers, the authors cited literature [33] to justify the use of the median (instead of the mean), arguing that the data distribution deviated from normality. However, in the data description (in text), they refer to the mean rather than the median (both values are presented in the tables, but there is no clear explanation of the distribution characteristics). If all the data were non-parametric, this should be explicitly stated, and the description should be adjusted accordingly, as it reveals the nature of the data. Otherwise, the authors are inconsistent and may mislead the reader regarding the interpretation of the values.

In all cases where nonparametric distribution data, the results should be described in the text using the median, and this should be clearly indicated. Analyzing the data range shows a high level of variability. Therefore, using mean values for comparisons in the text - instead of medians - is not appropriate when the distribution is clearly nonparametric.

“The study revealed that males, physically active individuals, and those classified as overweight or obese had higher water consumption. This can be explained by differences in metabolism and physiological needs. Men generally have a higher muscle mass than women, leading to increased water requirements for maintaining cellular hydration and metabolic processes [67]. Similarly, physically active individuals require more water to compensate for fluid lost through sweat during exercise, as well as to support enhanced metabolic activity during physical exertion [68]. Overweight and obese individuals often have higher metabolic rates and increased body mass, which necessitates greater fluid intake to regulate body temperature and maintain metabolic functions [69], [70].” - The statement regarding specific results ("Overweight and obese participants were more likely to drink water than underweight individuals") requires substantial correction. First, it is inappropriate to cite publications related to children (e.g., reference 70) when the study sample consists of adults – these are entirely incomparable datasets. Furthermore, references 69 and 70 actually suggest the opposite trend, indicating that obese individuals were less hydrated. The cited refrences does not indicate that they actually consume more water –  it only states that they should consume more due to inadequate hydration. Authors should refers to whether findings from other studies confirm that overweight and obese individuals consume more water than underweight individuals. The question concerns a methodological issue - whether there might be an error in this regard.

Author Response

Comments 1: Thank you very much to the authors for their responses. However, based on the replies received, I now have even more questions. These issues require clarification and most likely some corrections in the article.

In response to the question about the methodology used for the outliers, the authors cited literature [33] to justify the use of the median (instead of the mean), arguing that the data distribution deviated from normality. However, in the data description (in text), they refer to the mean rather than the median (both values are presented in the tables, but there is no clear explanation of the distribution characteristics). If all the data were non-parametric, this should be explicitly stated, and the description should be adjusted accordingly, as it reveals the nature of the data. Otherwise, the authors are inconsistent and may mislead the reader regarding the interpretation of the values.

In all cases where nonparametric distribution data, the results should be described in the text using the median, and this should be clearly indicated. Analyzing the data range shows a high level of variability. Therefore, using mean values for comparisons in the text - instead of medians - is not appropriate when the distribution is clearly nonparametric.

Response 1: Thank you very much for your valuable feedback and for giving us the opportunity to clarify our methodology.

We acknowledge the importance of accurately describing the data distribution and measures of central tendency. Following your suggestion, we have now explicitly stated in the Methods section that the distribution of all continuous variables was assessed using the Shapiro-Wilk test, which indicated that the data were not normally distributed and contained significant outliers. Therefore, median and range were chosen as the primary descriptive statistics to better represent the typical values in the skewed data. Mean and standard deviation values are also presented for reference but are no longer emphasized in the text to avoid any potential misinterpretation.

All relevant paragraphs in the Results section have been revised to consistently report medians when describing the central tendency of nonparametric data. Additionally, the table captions now specify that medians and ranges are the primary descriptive statistics, with means and standard deviations included for completeness.

Comments 2: “The study revealed that males, physically active individuals, and those classified as overweight or obese had higher water consumption. This can be explained by differences in metabolism and physiological needs. Men generally have a higher muscle mass than women, leading to increased water requirements for maintaining cellular hydration and metabolic processes [67]. Similarly, physically active individuals require more water to compensate for fluid lost through sweat during exercise, as well as to support enhanced metabolic activity during physical exertion [68]. Overweight and obese individuals often have higher metabolic rates and increased body mass, which necessitates greater fluid intake to regulate body temperature and maintain metabolic functions [69], [70].” –

The statement regarding specific results ("Overweight and obese participants were more likely to drink water than underweight individuals") requires substantial correction. First, it is inappropriate to cite publications related to children (e.g., reference 70) when the study sample consists of adults – these are entirely incomparable datasets. Furthermore, references 69 and 70 actually suggest the opposite trend, indicating that obese individuals were less hydrated. The cited refrences does not indicate that they actually consume more water –  it only states that they should consume more due to inadequate hydration. Authors should refers to whether findings from other studies confirm that overweight and obese individuals consume more water than underweight individuals. The question concerns a methodological issue - whether there might be an error in this regard..

Response 2: Thank you for your careful evaluation and for bringing this important issue to our attention. We agree with your observation that references 69 and 70 in the previous version were not appropriately aligned with the adult population in our study and did not directly support the claim that overweight and obese individuals consume more water than underweight individuals.

In response, we have revised the statement and replaced the previous references with more appropriate and population-relevant literature that focuses specifically on adult hydration behaviors and needs. The updated references now include studies that examine fluid intake patterns and hydration status among adults with varying BMI classifications.